# Profiles of histidine-rich glycoprotein associate with age and risk of all-cause mortality

Mun-Gwan Hong[1],[*] , Tea Dodig-Crnković[1],[*] , Xu Chen[2], Kimi Drobin[1], Woojoo Lee[2,3], Yunzhang Wang[2], Fredrik Edfors[1], David Kotol[1] , Cecilia Engel Thomas[1] , Ronald Sjöberg[1], Jacob Odeberg[1,4] , Anders Hamsten[5], Angela Silveira[5] , Per Hall[2,6] , Peter Nilsson[1] , Yudi Pawitan[2], Mathias Uhlén[1] , Nancy L Pedersen[2], Sara Hägg[2], Patrik KE Magnusson[2], Jochen M Schwenk[1]

Despite recognizing aging as a common risk factor of many human diseases, little is known about its molecular traits. To identify age-associated proteins circulating in human blood, we screened 156 individuals aged 50–92 using exploratory and multiplexed affinity proteomics assays. Profiling eight additional study sets (N = 3,987), performing antibody validation, and conducting a meta-analysis revealed a consistent age association ($P = 6.61 \times 10^{-6}$) for circulating histidine-rich glycoprotein (HRG). Sequence variants of HRG influenced how the protein was recognized in the immunoassays. Indeed, only the HRG profiles affected by rs9898 were associated with age and predicted the risk of mortality (HR = 1.25 per SD; 95% CI = 1.12–1.39; $P = 6.45 \times 10^{-5}$) during a follow-up period of 8.5 yr after blood sampling (IQR = 7.7–9.3 yr). Our affinity proteomics analysis found associations between the particular molecular traits of circulating HRG with age and all-cause mortality. The distinct profiles of this multipurpose protein could serve as an accessible and informative indicator of the physiological processes related to biological aging.

## Introduction

Aging is the single most dominant risk factor of common diseases in the elderly and of death in the human population (López-Otín et al, 2013). Molecular insights into aging could enable direct identification of future treatments for various diseases and would increase our understanding of longevity and related mechanisms. However, many of the underlying molecular processes and changes in humans remain poorly understood (López-Otín et al, 2013). Biological age or mortality risk have previously been investigated via DNA methylation, telomere length, proteomic studies, mining of clinical records (Ganna & Ingelsson, 2015; Jylhävä et al, 2017), and showed several candidates for these traits (Wiklund et al, 2010; Barron et al, 2015; Ganna & Ingelsson, 2015; Marioni et al, 2015).

There are currently two major technological concepts available for measuring the proteins circulating in blood-derived samples: affinity-based proteomics and mass spectrometry. Both approaches offer a unique window into human health and diseases and have been used to study subsets of nearly 5,000 proteins known to be circulating in blood (Schwenk et al, 2017). Affinity proteomics has initially suffered from a lack of binding reagents to the wider proteome, but antibody resources such as the Human Protein Atlas (HPA) (Uhlén et al, 2015) or aptamer-based platforms have enabled affinity proteomics for larger discovery projects, such as recently demonstrated in the context of aging (Lehallier et al, 2019). An important aspect for affinity proteomics is to validate the antibodies in a context-dependent manner (Uhlen et al, 2016) and using the power of population-based genome-wide association studies (GWAS) with circulating proteins (Suhre et al, 2017) can mitigate some of the uncertainty concerning target binding.

Using antibody assays based on suspension bead arrays (Byström et al, 2014), we profiled serum and plasma from a large number of individuals from different study sets. Studying the changes in plasma protein levels with age, we explored, filtered, and ranked plasma profiles associated with age across these sets of samples and confirmed antibody selectivity by genetic association tests and by applying different immunoassays (Fig S1).

## Results

We profiled the serum proteomes of 156 humans to screen for age-associated proteins that could serve as indicators of biological age.

[1]Department of Protein Science, Science for Life Laboratory, KTH–Royal Institute of Technology, Solna, Sweden    [2]Department of Medical Epidemiology and Biostatistics, Karolinska Institutet, Stockholm, Sweden    [3]Department of Public Health Science, Graduate School of Public Health, Seoul National University, Seoul, Korea    [4]Department of Medicine Solna, Karolinska Institutet and Karolinska University Hospital, Solna, Sweden    [5]Department of Medicine Solna, Cardiovascular Medicine Unit, Karolinska Institutet, Solna, Sweden    [6]Department of Oncology, Södersjukhuset, Stockholm, Sweden

Correspondence: jochen.schwenk@scilifelab.se
*Mun-Gwan Hong and Tea Dodig-Crnković contributed equally to this work

The most significant finding was further investigated in 3,987 additional samples from eight different study sets (Table 1). An approach using different experimental methods and genomic data was used to validate antibody binding. The protein profiles were examined in relation to several clinical traits and tested as predictors of mortality risk, possibly reflecting biological aging.

## Screening for age-associated profiles

Age-associated protein profiles were first investigated in a set of 156 human subjects selected in age intervals of 5 yr from a Swedish twin cohort (denoted set 1, summarized in Table 1). The gender-matched samples included 30 monozygotic (MZ) twin pairs, who were 50–70 yr old. Assays using a total of 7,258 HPA antibodies were applied to profile age-associated proteins in serum. The average intraclass correlation (ICC) within twin pairs of antibody profiles was weak (ICC = 0.26). Minimal remaining effects of the twin relationship were corroborated by a linear mixed model that considered the dependency.

For this screening, the targets were purely determined by the availability of antibodies. The set of antibodies comprised targets from 6,370 protein-encoding genes (about 32% of the non-redundant human proteome) and profiles were obtained using antibody suspension bead array assays. The acquired data were preprocessed and quality controlled, which included outlier removal and normalization to account for experimental variation across assay plates and data batches (details in the Materials and Methods section). Linear regression models (LM) then determined the protein profiles that changed monotonically with increasing age. The models revealed only one of 7,258 protein profiles as significantly age-associated (at $\alpha$ 0.01, Table S1), when screening the sera of individuals in set one (adjusted $P$ = 4.69 × $10^{-5}$). The association was also significant in the model considering twin-pair dependency (adj. $P$ = 8.62 × $10^{-5}$).

## Replication of the discovered age associations

Next, we continued to study the top finding in eight additional sample sets (set 2–9, Table 1). Details about the additional cohorts and sample selection are provided in the Supplemental Data 1 (see Samples and selections). We found consistent age-associated trends with the antibody HPA045005 across all eight replication sets (Fig S2). The combined effect of age on HPA045005 in all 9 sets excluding 45 overlapping samples (details in Supplemental Data 1) was estimated using a random effects model accounting for differences in age ranges and distribution, showing a significant association between HPA045005 profile and age (meta-analysis, $P$ = 6.61 × $10^{-6}$, Fig 1).

We then investigated the connection between genetic data and protein profiles obtained by HPA045005. A possible association in *cis* could give information about the circulating proteins captured by the antibody in our assays. Using GWAS to ~8.8M genetic variants imputed from >700K single nucleotide variants genotyped by Illumina BeadChip in sample set 3 (N = 2,308), we identified a single locus in chromosome 3q27.3 to be associated with the antibody profile (adj. $P$ < 0.01, Fig 2A). The locus spans two genes, *FETUB* and *HRG* (Fig 2B). The most significantly associated genetic variant in the locus was the single nucleotide polymorphism (SNP) rs9898 ($P$ = 2.35 × $10^{-97}$, minor allele frequency [MAF] = 0.32). This SNP leads to an amino acid change in the sequence of histidine-rich glycoprotein (HRG) from Pro204 to Ser204 (pro-form). Three additional markers in the peak (rs1042464, rs2228243, and rs10770) are non-synonymous and two SNPs (rs3890864 and rs56376528) are located near (<2 kbp) to the transcription start site (Table S2). All are located in exons or upstream of *HRG*. This GWAS result indicates that the antibody detected HRG, which is an abundant blood protein secreted by the liver (Uhlén et al, 2019). Associations of plasma HRG levels to SNPs, including rs2228243, have been found in previous plasma profiling studies (Suhre et al, 2017).

## Validation and annotation of HRG profiles

Next, we confirmed the binding selectivity of HPA045005 to HRG by first using a bead-based sandwich immunoassay. Beads carrying HPA045005, another anti-HRG antibody (HPA054598), as well as negative controls were combined to detect full-length recombinant

**Table 1. Description of sample sets.**

| Study set | Age (yr) | Gender (F:M) | Indication[a] | Cohort name | Sample Type | References |
|---|---|---|---|---|---|---|
| Set 1 | 50–92 | 78:78 | Population | TwinGene | Serum | Lichtenstein et al (2002) and Magnusson et al (2013) |
| Set 2 | 9–63 | 102:102 | Population | LifeGene | Plasma | Almqvist et al (2011) |
| Set 3 | 48–93 | 1,613:1,386 | Population | TwinGene | Serum | Lichtenstein et al (2002) and Magnusson et al (2013) |
| Set 4 | 51–86 | 50:0 | Breast cancer | | | |
| Set 5 | 56–75 | 0:50 | Prostate cancer | | | |
| Set 6 | 55–78 | 16:27 | Cardiovascular disease | IMPROVE | Plasma | Baldassarre et al (2010) |
| Set 7 | 41–60 | 12:31 | Myocardial infarction | SCARF | Plasma | Samnegård et al (2005) |
| Set 8 | 48–73 | 20:23 | Acute coronary heart syndrome | CHAPS | Plasma | Odeberg et al (2014) |
| Set 9 | 40–73 | 600:0 | Mammography | KARMA | Plasma | Gabrielson et al (2017) |

[a]Subjects included in the presented study did not include individuals diagnosed with the disease of the indication area, but the subjects assigned as controls for the different disease cohorts.

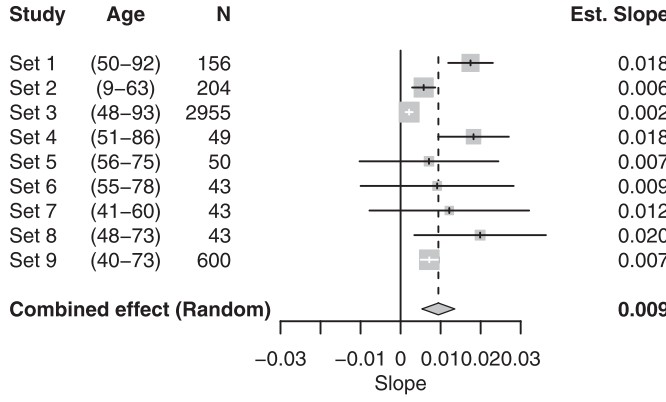

**Figure 1. Meta-analysis from nine different sample sets.**
In the forest plot, the numbers in parenthesis indicate the age range of the included subjects. For each sample set, the estimated effect of age on HPA045005-derived profiles from the linear regression model, 95% confidence interval of it, and study weight in the meta-analysis are shown in the middle as a tick, a line, and a gray box, respectively. The numeric value of the effect is clarified at the right side.

HRG in serial dilution. We found that pairing both HPA045005 and HPA054598 with a biotinylated version of HPA054598 allowed us to detect HRG in a concentration-dependent manner (Fig S3). Here, the curves obtained from both antibody pairs were substantially different and higher than the internal negative controls.

To elucidate the binding selectivity of HPA045005 against other antigens, we applied the antibody to a large protein microarray. Among >10,000 antigens, the antibody exclusively bound to its corresponding antigen (Fig S3), which indicated that the antibody does not generally cross-react with other human antigens in an unspecific manner. We also aimed at determining the binding of HPA045005 to peptides representing its antigen using high-density peptide arrays. The antigen did not show a significant recognition of these peptides above background (data not shown). Hence, binding analysis supports the findings from GWAS that HPA045005 captures HRG from serum and plasma in the single binder assay.

In addition to HPA045005, GWAS was performed for a monoclonal binder targeted the HRG protein (BSI0137) with sample set 3. For BSI0137, there was one locus in the gene *HRG* which was strongly associated with the antibody's profile (top pQTL was associated with $P < 1 \times 10^{-300}$, Fig S4). Interestingly, the identified locus included all the four non-synonymous SNPs previously observed to associate with HPA045005. However, the most significant SNP was not rs9898 but rs1042464, and the slopes of correlating BSI0137 with these SNPs were opposed to those for HPA045005 (Fig 2C). Applying Probabilistic Identification of Causal SNPs (PICS) (Farh et al, 2015), we confirmed that it was highly unlikely to observe rs9898 as the most significantly associated SNP with HPA045005 when rs1042464 was the causal SNP (none in 100,000 permutations), whereas the significance of rs1042464 was within possible range assuming rs9898 was causal (Fig S5). The distance from the mean of the permuted *P*-values after log-transformation was about 1.22 SD of the values. Likewise, the PICS applied for BSI0137 demonstrated that rs1042464 was much more likely to be causal than rs9898 for this

antibody profile (Fig S5). The rs9898 causes HRG to contain either Pro204 or Ser204. Profiles of circulating HRG obtained from HPA045005 increased with the number of major allele *C*, which can produce only Pro204 form, in a dosage-dependent manner. Profiles of HRG reported by BSI0137 increased with the number of *T*-alleles of rs1042464 for Ile493. Hence, the PICS analysis revealed a selective binding affinity of HPA045005 toward another site of HRG than BSI0137. For these two binders, this points at differential selective affinities toward HRG variants: HPA045005 has a preference for HRG with Pro204 over Ser204 compared with BSI0137, which prefers HRG with Ile493 over Asn493 (Table 2). The variants are Pro204Ser and Ile493Asn and located in two separate domains of HRG, as illustrated in Fig 2D, and each variant influenced one of the two distinct protein profiles.

Next, we investigated the associations of the HRG profiles obtained by the two distinct binders, to 10 clinical traits available in sample set 3. We applied a linear model that included the top SNP of each binder to account for the differences between the genotypes. For HPA045005-derived HRG profiles, we observed negative associations to blood levels of hemoglobin (Hb), apolipoprotein A1 (APOA1), triglycerides (TG), and total cholesterol (TC), as well as a positive association to C-reactive protein (CRP). For BSI0137-derived profiles, there were significant associations to traits reflecting lipid metabolism such as levels of apolipoprotein B (APOB), low-density lipoprotein, TC, TG, and expected negative associations to high-density lipoprotein. The results summarized in Table 2 show that different molecular traits may be associated to HRG depending on which binder and epitope was used. In addition, we checked the association between the HRG profiles from HPA045005 and activated partial thromboplastin time (aPTT) in set 8. It was found negatively correlated with statistical significance (R = −0.55, *P* = 0.05).

Last, mass spectrometry analysis of serum was carried out by LC–MS/MS to report peptides related to the HRG and to search for those representing variants rs9898, rs10770, rs2228243, and rs1042464. HRG peptides representing 96.5% of the sequence have been reported on PeptideAtlas (Desiere et al, 2006). HRG protein is an abundant blood protein of 30–100 μg/ml (Schwenk et al, 2017), so no pre-fractionation or depletion of abundant proteins was performed prior analysis. One pool of serum was digested using five different proteases (Table S3) to increase the possibility that variant related peptides could be identified. As shown in Fig S6, the detected peptides represented the region of rs1042464 (Asn493 and Ile493). In addition, data from the PeptideAtlas was used to search for evidence of the SNPs, and peptide representing rs9898 were only found once as compared with >2,800 times for rs1042464. Peptides referring to rs10770 and rs2228243 were, however, not detected and peptides related to rs9898 were very unlikely to be observed because of the altered amino acid composition around the cleavage site, resulting in peptides of unsuitable peptide lengths. In our study, we could not obtain MS data to support the observed differential protein profiles.

## Mortality association and prediction

Finding the association of HRG and age led us to further study biological age in relation to mortality. We accessed the Swedish

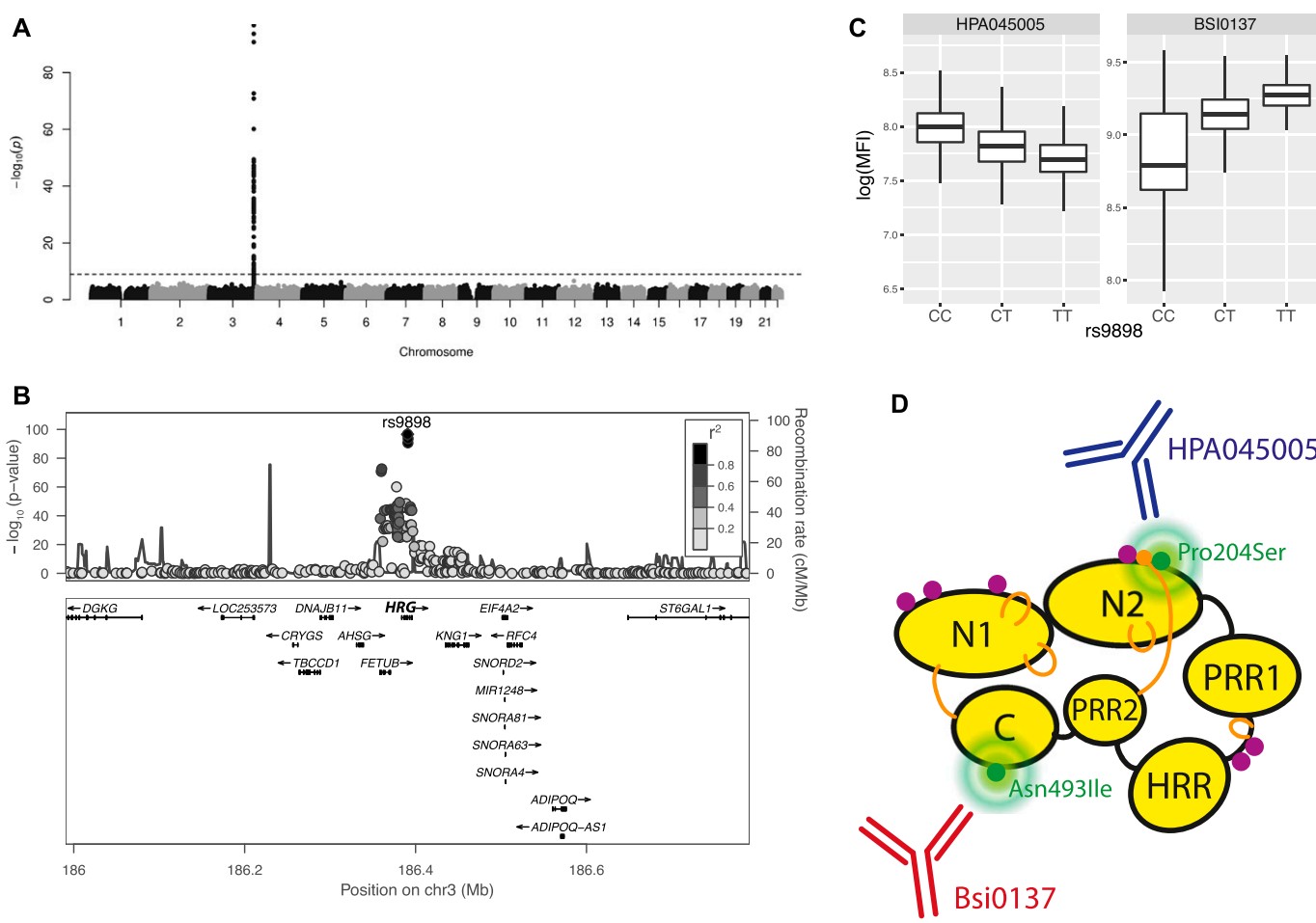

**Figure 2. Genome-wide association study results and histidine-rich glycoprotein (HRG) domains.**
**(A)** Manhattan plot. The significance of association between genotypes and HPA045005 profiles is presented vertically. The dashed guide line marks the stringent threshold of *P*-value for genome-wide association study, which is *P* = 0.01 after Bonferroni correction. One peak in chromosome 3 indicates strong association of a locus with the molecular phenotype. **(B)** LocusZoom (Pruim et al, 2010) on associated locus. The illustration shows the elements of chromosome 3 associated with HPA045005 profiles. **(A)** Zooming in on the peak of the Manhattan plot in (A), the genes around the locus are presented together with the associated single-nucleotide polymorphisms. **(C)** Box plots to show the association between genotypes of rs9898 and two antibody profiles, HPA045005 and BSI0137. The trends were opposite. **(D)** Representation of HRG to illustrate how two separate domains of the HRG protein affected the profiles of the antibodies. Protein domains, glycosylation sites and disulfide bonds of HRG were schematically illustrated using black round, purple dots and yellow lines, respectively. The figure was prepared based on the schematic representation of Poon et al (2011).

death registry for information on whether the subjects were still alive or not within the follow-up time of ~8.5 yr (IQR = 7.7–9.3). We chose the largest sample set of the subjects at mid to old ages that spanned the average life expectancy in Sweden, which was sample set 3 (N = 2,973, 48–93 yr old). This was to gain statistical power needed for the analysis of all-cause mortality, which was otherwise a relatively rare event. A Cox proportional hazards model with age as the time scale was used with the adjustment for the effects of gender. This revealed that the profiles from HRG obtained by HPA045005 were significantly associated with the mortality risk during follow-up ($P$ = 1.13 × $10^{-4}$). In contrast, the HRG profiles determined by BSI0137, which were not correlated with those of the HPA045005 ($R^2$ = 0.006), were not associated to mortality ($P$ = 0.57).

To adjust for the effect of chronological age at sampling on the mortality association of the HPA045005-derived HRG profiles, the data were standardized using a linear model for age and age squared for each gender separately. The hazards model using the standardized HPA045005-HRG value affirmed the association with mortality ($P$ = 6.45 × $10^{-5}$). The risk of all-cause mortality was estimated to increase 1.25 times per SD of the age- and gender-adjusted HRG values (95% confidence interval = 1.12–1.39). In the model accounting for potential genetic effect of the most significantly associated SNP rs9898, the estimated hazard ratio (HR) of HRG became even higher with similar significance (HR = 1.31 per SD, $P$ = 7.75 × $10^{-5}$, N = 2,307). No evident difference of HR and average age was observed when stratifying by the genotype of the SNP on HR of HRG (Table S4). By examining the cause of death data, higher HPA045005-HRG profiles were found to elevate mortality risk by diseases of the circulatory system (HR = 1.46 per SD, $P$ = 2.80 × $10^{-4}$, ICD-10 code I00-I99) to a larger extent than by malignant neoplasms (HR = 1.28 per SD, $P$ = 1.73 × $10^{-2}$, ICD-10 code C00-C97) or other causes (HR = 1.20 per SD, $P$ = 0.25, Table S6).

**Table 2. Associations of histidine-rich glycoprotein profiles to various traits in set 3.**

| Trait | HPA045005 | | Bsi0137 | |
|---|---|---|---|---|
| | Estimate[a] | P-value | Estimate[a] | P-value |
| Aging | | | | |
| Age | 0.010 | $1.53 \times 10^{-6}$ | 0.0017 | 0.44 |
| Mortality risk | 1.25 | $6.45 \times 10^{-5}$ | 1.03 | 0.59 |
| Genetic/protein variants | | | | |
| rs9898 (Pro204Ser) | 0.15 | $2.35 \times 10^{-97}$ | −0.23 | $1.85 \times 10^{-177}$ |
| rs1042464 (Asn493Ile) | 0.10 | $1.9 \times 10^{-44}$ | −0.33 | $<1 \times 10^{-300}$ |
| Clinical trait | | | | |
| APOA1 | −0.26 | $9.47 \times 10^{-6}$ | −0.01 | 0.76 |
| APOB | −0.23 | $8.46 \times 10^{-4}$ | 0.30 | $6.81 \times 10^{-10}$ |
| C-reactive protein | 0.012 | $7.33 \times 10^{-5}$ | 0.0003 | 0.88 |
| Glucose | −0.028 | 0.061 | 0.013 | 0.21 |
| Hb | −0.008 | $6.45 \times 10^{-9}$ | 0.0004 | 0.68 |
| HbA1C | −0.047 | 0.070 | 0.030 | 0.092 |
| High density lipoprotein | −0.035 | 0.41 | −0.120 | $5.65 \times 10^{-5}$ |
| Low density lipoprotein | −0.044 | $1.53 \times 10^{-2}$ | 0.071 | $1.90 \times 10^{-8}$ |
| TC | −0.060 | $1.37 \times 10^{-4}$ | 0.060 | $4.92 \times 10^{-8}$ |
| TG | −0.086 | $2.73 \times 10^{-5}$ | 0.074 | $2.36 \times 10^{-7}$ |

[a]The estimates from selected models for individual associations. For clinical traits, the values are the estimated slope from linear regression models with adjustment for age and the top single-nucleotide polymorphism (rs9898 for HPA045005 and rs1042464 for BSI0137). Linear models were also used for age and genetic/protein variants, whereas Cox models for mortality risk (more details in the Materials and Methods section). For the trait, hazard ratios are presented in the column.

The two distinct histidin-rich glycoprotein profiles were compared with respect to the association with various traits in set 3, which are 2,999 samples from the TwinGene cohort (Lichtenstein et al, 2002; Magnusson et al, 2013).

A Cox model stratified by gender suggested somewhat stronger mortality association in women (HR = 1.35 per SD, $P = 2.13 \times 10^{-4}$) than in men (HR = 1.15 per SD, $P = 0.059$). Comparing extreme subsets with standardized HPA045005-HRG in the upper and lower quartiles demonstrated that the difference of median age at death was 1.8 yr in favor of the bottom quartile ($P = 3.87 \times 10^{-3}$, HR = 1.54, Fig 3). The difference was 1.9 yr in men (86.9 yr versus 85.0) and 0.6 yr in women (89.6 versus 89.0; Fig S7). The difference in life expectancy between the two extreme quartiles at the age of 45 was 3.7 yr in women (87.3 versus 91.0) and 2.8 yr in men (83.9 versus 86.7), assuming age-at-death follows a Weibull distribution (Table S5).

A comparison with the potential influence of inflammation on survival was done by using clinically measured CRP. As for HPA045005-HRG, two Cox models were fitted using 1) CRP and 2) age-adjusted CRP levels. The latter was obtained using the same linear model as HPA045005-HRG and adjusting for the same covariate. Associations of CRP model 1 (HR = 1.07 per SD, $P = 0.023$) and model 2 (HR = 1.01, $P = 0.023$) were less pronounced than for HPA045005-HRG. Next, we included CRP as a covariate in the Cox model for HPA045005-HRG to determine if inflammation in general would have an influence of HPA045005-HRG-related mortality. Negligible attenuation of the HPA045005-HRG association was observed after additional adjustment for CRP (HR = 1.25–1.24 per SD, $P = 6.45 \times 10^{-5}$ to $P = 1.05 \times 10^{-4}$). We also assessed the relation of HPA045005-HRG profiles to age and mortality considering diabetes-

related traits to account for any short-term effects of glucose. The results of the Cox models individually adjusted for nine additional clinical variables listed in Table 2 were barely changed for all models (Table S6). For each of the survival analyses, we confirmed that none of the hazard models violated the proportionality assumption of the Cox model using Schoenfeld residuals (Grambsch & Therneau, 1994). A summary of the survival analyses with 95% confidence interval is presented in Table S6.

## Discussion

### HRG is a multifunctional protein circulating in blood

We analyzed the relation between age and protein profiles in blood by multiplexed antibody-based assays and found a consistent, positive association with the antibody HPA045005. Validating the binding of the antibody using GWAS, protein microarrays and sandwich assays revealed that HRG was captured from serum and plasma. We further demonstrated that a selective binding affinity toward a domain of HRG revealed associations with mortality risk, unlike another anti-HRG antibody affected by a different HRG domain.

HRG has been described as an abundant protein in human blood plasma, and according to mRNA sequencing data of human tissues,

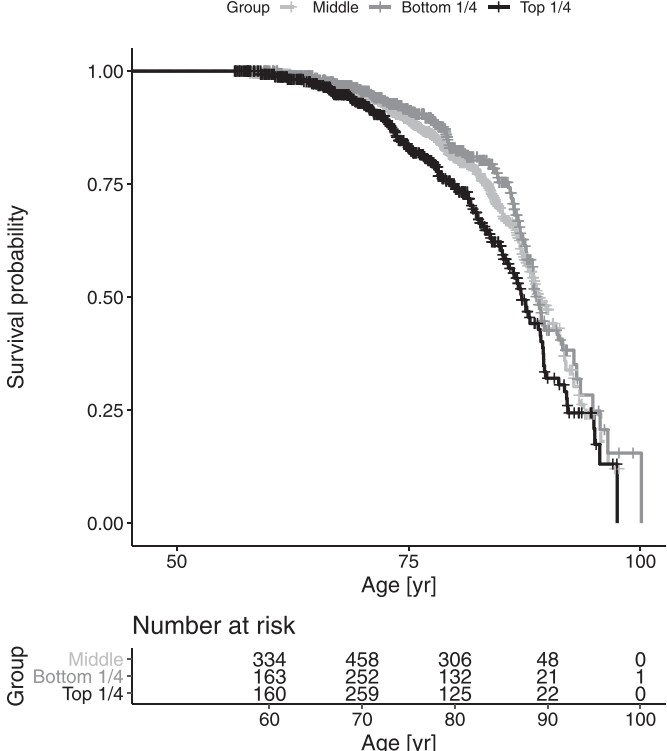

**Figure 3. Survival analysis comparing upper and lower quarters of histidine-rich glycoprotein levels.**

The individuals of sample set 3 were divided into four subsets by the quartiles of histidine-rich glycoprotein levels. Differential mortality across follow-up time is illustrated by the survival curves, where age was used as the time scale (Thiébaut & Bénichou, 2004). The number at risk at 10 yr intervals were displayed below the survival curve. Some detailed statistics related to this survival analysis are presented in Table S4.

HRG is exclusively expressed in the liver (Uhlén et al, 2015, 2019). The protein has been characterized to interact with diverse molecules, including heparin, heme, immunoglobulin G, $Zn^{2+}$, and complement components (Poon et al, 2011; Priebatsch et al, 2017), and particular functions have been assigned to each of its six domains (Martin et al, 2018). HRG is involved not only in immune response toward foreign substances and clearance of dead cells but also in vascular biology including anti-coagulation (Poon et al, 2011). HRG levels have previously been correlated and linked to blood ABO type and age (Drasin & Sahud, 1996). The protein has been named a bio-marker of preeclampsia, which entails angiogenic imbalance and defective coagulation control (Bolin et al, 2011), and found as a marker for sepsis prediction among systemic inflammatory response syndrome patients (Kuroda et al, 2018). Partly because of its molecular composition and abundance, HRG has been assigned to many other biological processes. When searching for possible protein–protein interactions of HRG using the STRING database (Szklarczyk et al, 2019), we observed that the number of listed interactions were enriched for platelet degranulation (Gene Ontology: 0002576). Many of the proteins in the HRG-interactome were also expressed by the liver and secreted into blood (Uhlén et al, 2019).

With the current data, it remains difficult to determine the most plausible mechanism by which HRG profiles are increased in the process of aging and why this observation was only seen for one of the anti-HRG antibodies but not for the other. However, HRG has functional similarities with CRP, which is another indicator of aging and mortality (Barron et al, 2015). Their related functions in the context of coagulation and inflammation (Poon et al, 2011) may increase the likelihood that both change with age because of similar reasons. Interestingly, we found a negative association of the HRG profiles of HPA045005 to levels of blood hemoglobin, which could otherwise point at the involvement of this HRG in binding free heme from erythrocytes. Observing a negative correlation between the HPA045005-HRG profiles and aPTT, for which a prolonged time indicates lower thrombosis risk, supports a hematological hypothesis. Lower HPA045005-HRG profile might indicate lower risk of thrombosis, and thereby also mortality. In contrast, the BSI0137 profiles of HRG affected by variants on residue 493 did reveal links to lipid metabolism but no significant association to Hb levels. This supports the hypothesis that a site- and variant-specific binding of the two antibodies (via distinct epitopes) reveal different perspectives about the traits of HRG.

In the *HRG* gene, there are four relatively common genetic variants (MAF > 10%) that lead to amino acid polymorphisms. In a genetic study investigating the aPTT, Houlihan et al (2010) observed that the minor *T* allele of rs9898 was associated with shorter aPTT, suggesting an elevated risk of thrombosis in these individuals (Houlihan et al, 2010). Houlihan et al (2010) proposed a potential interaction between HRG and thrombosis; hence, thrombosis could be a possible mediator between elevated HRG and risk of mortality. The genetic association seems opposite to the two correlations we observed here, 1) more *T* allele and lower HRG profiles of HPA045005 and 2) lower HPA045005-HRG profiles and longer aPTT. However, considering that the correlation of the HPA045005-HRG profiles with rs9898 was mainly driven by molecular characteristic of our immunoassay, the opposite trend strengthened our hypothesis. The HPA045005-HRG profiles were possibly associated with mortality not through rs9898 but thrombosis. Tang et al (2012) confirmed the observations of Houlihan et al (2010) and pointed out that possible interactions between causal variants of *HRG*, *KNG1*, and *F12* may further influence coagulation and aPTT (Tang et al, 2012).

We also performed a look-up on the variants using the data hosted by the Genotype-Tissue Expression (GTEx) Project portal (version 8, accessed 2019-11-19). We searched for any significant expression QTL or splice QTL that could provide further insights into circulating levels of the secreted HRG. For the rs9898 (as well as rs2228243 and rs10770), there were, however, neither any expression QTLs nor splice QTLs reported in GTEx. The rs9898 does not alter the mRNA expression or is associated with the formation of splice variants of *HRG*.

GWAS analysis revealed the associations of both antibodies' profiles with non-synonymous SNPs. The PICS analysis indicated which amino acid residue of the HRG may affect the antibody recognition. Using this novel approach for identifying recognition sites, we found that HRG profiles from HPA045005 differentiated between HRG variants with Pro204 over Ser204. This might imply that the antibody had its HRG-binding epitope in proximity to residue 204, as it was predicted to be exposed (Fig S8). This position is located in HRG's cysteine protease inhibitor domain for which a

lacking inhibitory activity was previously described (Ochieng & Chaudhuri, 2010). Interestingly, two of the neighboring residues to Ser204 or Pro204 are amino acids with functional groups (Fig 2D): Asn202 is an N-glycosylation site found for Ser204 variants (Hennis et al, 1995) and Cys203 has been identified as a site for glutathione modification (Kassaar et al, 2014). This may imply that any changes around residue 204 can affect molecular functions of the protein. It is noteworthy that no single genetic variant around *HRG* reached genome-wide significance for mortality risk in a study including the TwinGene cohort (Ganna et al, 2013). Together with the contrasting difference we observed of the HRG profiles obtained by the other antibody, we postulate that molecular alteration around the 204[th] residue was associated with age and mortality risk. Investigating changes at the spot may assist in further categorizing HRG's involvement in a diverse set of physiological processes.

## Limitation and variation

Circulating HRG was profiled across ages and samples from different donors to cover a broad range of lifespans. Even though finding consistent trends of HRG in serum and plasma of the cross-sectional study, the performance of the HRG assays can be influenced by the type of blood preparation. This could influence the degree to which HRG associated with age. On the other hand, gradual associations were repeatedly observed in multiple independent study sets derived from different Swedish cohorts, which provided supportive evidence about the association of the HPA045005-HRG with age. As HPA045005 profiles of circulating HRG increased as age advances, an analysis of longitudinal samples collected from representative subjects could be a viable strategy. As we also found that the elevated HPA04500-HRG was correlated with higher risk of mortality even after adjusting for genetic effects of the strongly associated SNP, the age-dependent profile may imply a time-wise transition along individual ages, possibly biological ages. Last, molecular investigations on the residues, including the 204[th] amino acid, are needed to further interpret the increasing trends of the HPA045005-HRG profiles, and as suggested by the differential associations to the clinical traits.

With increasing knowledge about genetic effects on the circulating proteome comes the challenge to validate these observations. As we have seen, genetic variation can cause two antibodies to reveal discordant protein proteins even though they bind the same protein. We acknowledge the limitation in our results to further study the effects on the epitopes of HRG such as by applying additional antibodies and establishing appropriate methods. Such tools would further strengthen the validity and better enable others to reproduce the made observations.

We observed variation in the degree to which HRG associated with age, which is visible in Fig 1. To some extent, the variation can be explained by technical limitations resulting in different signal intensity ranges determined in each study set. The method was primarily developed to screen larger numbers of protein profiles for possible associations to different traits, hence it is tailored toward shortlisting of candidates rather than standardized to determine absolute abundance levels. Seeing that the estimated slopes from sample sets 2, 3, and 9 were relatively lower than the values from the other sets, some parts of the variation might originate from the

difference in age range and sample source, sample collection and preparation, or selection of participants. For example, the individuals in the sample sets 2, 7, and 9 were substantially younger (median age 40, 52, and 54 yr, respectively) compared with all others (~65 yr old). The sample sets 2, 3, and 9 were near population-based, whereas the others were healthy individuals except those in sample set 1, in which older women and men were overrepresented because of same number of individuals selected per age-group.

## HRG as a potential predictor of mortality

Several other molecular indicators have been previously reported to predict mortality risk. Barron et al (2015) showed that CRP (HR = 1.42), N-terminal pro brain natriuretic peptide (NT-proBNP, HR = 1.43), and white blood cell count (HR = 1.36) were statistically significant in meta-analyses (Barron et al, 2015). The HR estimate of HRG in our study (1.53 between top and bottom quarters) was comparable. Schnabel et al (2013) linked CRP to mortality risk in a follow-up period of median 8.9 yr (Schnabel et al, 2013). Using HR per SD, their estimate for CRP was 1.18, which was similar to our estimate of 1.25 from HRG. HR per SD of DNA methylation (1.09–1.21) in the study by Marioni et al (2015) was also comparable to our HRG estimate (Marioni et al, 2015). Ganna and Ingelsson (2015) used questionnaire-derived measures for an extensive population-based mortality study (Ganna & Ingelsson, 2015). Their top predictors resulted in Harrell's C-index = 0.74 when including age, and using the same model, HRG performed in the same range (C-index = 0.77).

Indeed, other proteins such as growth differentiation factor 15 (GDF15), IL-6, and CRP have previously been described in aging and all-cause mortality independent of telomere length (Wiklund et al, 2010). Other recent large-scale affinity proteomics approaches have reported age associations of circulating proteins, for example, GDF15 but also other proteins of the coagulation system (Tanaka et al, 2018) and heparin binding function (Lehallier et al, 2019). Although HRG is abundant in blood (Poon et al, 2011), those studies did not measure the protein, reflecting challenges in assay development. It is noteworthy that circulating GDF15 can also be highly influenced by medications such as metformin (Gerstein et al, 2017) and the expression of the *GDF15* gene, also known as nonsteroidal anti-inflammatory drugs-activated gene (*NAG-1*), can be induced by other common drugs (Wang et al, 2011). We acknowledge that a better understanding of HRG in the context of aging and mortality will require more lifestyle data from the donors. Last, our presented strategy was of an exploratory nature and we did not actively include antibodies toward previously known age-related proteins, such as GDF15. The shortlisted targets represented those for which the antibodies performed in the antibody bead array method when screening serum and plasma for indicators of aging.

In conclusion, we have described distinct profiles of circulating HRG as indicator of aging and mortality. Extensive efforts were put into confirming our observation across independent cohorts and applying molecular approaches to characterize the differential recognition of HRG. As a known multipurpose adapter protein, HRG plays a role in hemostasis, and the profiles of the protein can serve as a predictive indicator for all-cause mortality within 8.5 yr after blood draw; hence, the particular profiles of circulating HRG could

be helpful as an accessible indicator of processes linked to biological aging.

# Materials and Methods

**Cohort design and sample selection**

### Sample set 1 from TwinGene cohort

A population-wide collection of blood from 12,614 twins born between 1911 and 1958 has been undertaken in a project named TwinGene. The primary aim of the TwinGene project has been to systematically transform the oldest cohorts of the Swedish Twin Registry (STR) into a molecular-genetic resource (Magnusson et al, 2013). From 2004 to 2008, a total of 21,500 twins (~200 twin pairs per month) were contacted by invitation to the study and were provided information of the study and its purpose, consent forms, and health questionnaire. The study population was limited to those participating in the Screening Across the Lifespan Twin Study (SALT) which was a telephone interview study conducted in 1998–2002 (Lichtenstein et al, 2002). Other inclusion criteria were that both twins in the pair had to be alive and living in Sweden. Subjects were excluded from the study who had declined to participate in future studies or had been enrolled in other STR DNA sampling projects. When the signed consent forms returned, blood-sampling equipment was sent to the subjects, who were asked to visit local healthcare facilities in the morning, after fasting from 20:00 the previous night, from Monday to Thursday and not the day before a national holiday. This was to ensure that the sample tube would be delivered to the Karolinska Institutet Biobank by the following morning by overnight mail. After arrival, the serum was stored in liquid nitrogen.

The contribution for sample set 1 of serum samples from the TwinGene study consisted of: (i) samples from 96 unrelated twins distributed in groups of 12 subjects (6 males and 6 females) in each age strata 50, 55, 60, 65, 70, 75, 80, and 85 yr of age, and (ii) samples from 60 MZ twins (30 complete pairs) distributed in groups of 12 (3 male pairs and 3 female pairs) in each age strata of 50, 55, 60, 65, and 70 yr of age. The width of the age intervals was approximately ±3 mo.

### Sample set 2 from LifeGene cohort

LifeGene is a prospective cohort study that includes collection of plasma and serum, tests of physical performance, as well as questionnaire responses regarding a wide range of lifestyle factors, health behaviors, and symptoms (Almqvist et al, 2011). Participants responded to a web-based questionnaire and book time for a visit to a LifeGene test center, at which blood samples are taken. EDTA plasma was processed at the test center as follows: the EDTA tube with a gel plug was centrifuged, put into –20°C before shipment in a cold chain. All samples were sent to Karolinska Institutet Biobank for further separation into aliquots in REMP plates and frozen at –70°C. All participants or, in the case of children under the age of 11, their guardians, provided signed consent.

The sample set 2 cohort consisted of five male and five female samples randomly chosen from each of the ages <5, 10, 15, 20, 25, 30, 35, 40, 45, 50, and 55 (±3 mo). For 12 participants, serum was also available.

### Sample sets 3–5 from TwinGene cohort

Sample sets 3, 4, and 5 were selected from the same cohort, TwinGene (Magnusson et al, 2013), as sample set 1 (described above). Of 132 microtiter 96-well microtiter plates for storage of TwinGene samples, the 12 plates having the largest age span (>20 yr) among samples in a plate and another randomly chosen 20 plates comprising a sufficient number of samples (>91) were selected. Sample set 3 consisted of the 3,000 samples in the selected 32 storage places. The data of one individual were removed in the analyses because age of the subject was missing. Independently from the sample selection, sample sets 4 and 5 were age- and gender-matched controls for breast and prostate cancer studies, respectively. The mortality data were obtained by linkage of individuals in TwinGene to the data in the Swedish tax authorities by personal identification number. The data were updated on 10 January, 2015. Likewise, the ICD-10 data from the Swedish cause of death register were obtained, which was updated on 31 December, 2012. Clinical blood chemistry assessments were performed by the Karolinska University Laboratory for the following biomarkers: total cholesterol, triglycerides, high-density lipoprotein, low-density lipoprotein (by Friedewald formula), CRP, glucose, APOA1, APOB, Hb, and HbA1c (Magnusson et al, 2013). Clinical blood chemistry assessments were performed by the Karolinska University Hospital Laboratory. Levels of HbA1c were measured by a high-liquid performance chromatography separation technique. Levels of the other biomarkers were determined by Synchron LX systems (Beckman Coulter) (Rahman et al, 2009).

### Sample sets 6–9

The sample sets 6–9 are described in Supplemental Data 1.

### Ethics

All the studies were approved by the Ethics Board of the corresponding hospital or institution and conducted in agreement with the Declaration of Helsinki. The ethical approval document numbers are 2007/644-31/2 for TwinGene, 2009/615-31/1 for LifeGene, 03-115 and 2017/404-32 for IMPROVE, 95-397 and 02-091 for SCARF, EPN 2009/762 and LU 298-91 for CHAPS, and 2010/958-31/1 for Karma. All subjects, or their guardians, provided their informed consent to participation in the individual studies.

**Plasma proteomics analysis methods**

Experimental details on the molecular methods and data processing are available as Supplemental Data 1. This includes descriptions for the antibody selection, bead array assay, sandwich immunoassays, mass spectrometry, and protein microarray analysis. Because of substantial difference between the composition of the two sample types (Fig S9), the data of plasma and serum were processed separately.

**GWAS**

Genomic DNA from all available dizygotic twins and one member of each monozygotic twin pair were genotyped by using Illumina OmniExpress BeadChip (700K). Genotyping QC exclusion criteria: genotypic or individual missingness > 0.03, MAF < 0.01, Hardy-

Weinberg equilibrium $P < 10^{-7}$, gender mismatch, heterozygosity (individuals with an F-statistic beyond ±5 SD from the sample mean), or cryptic relatedness. The 1000 Genome reference panel (GRCh37/hg19, Phase 1, version 3) was used for imputation, by using Mach 1.0 and Minimac. After genotype antibody-profile match, GWAS was performed among 2,308 twins by using PLINK 1.90 beta. Analyses were restricted to autosomal SNPs with imputation quality (info or $r^2$) >0.4. The first four principal components were included to control population stratification. The "within" option in PLINK was used to statistically adjust for relatedness (complete dizygotic twin pairs). Manhattan plots were drawn using qqman package in R 3.4.1. The mutation types of the associated SNPs were obtained from UCSC Genome Browser (https://genome.ucsc.edu) using human "GRCh38/hg38" assembly and "snp150Common" (dbSNP build 150, ≥1% MAF) table, which was accessed on 20 September, 2018.

### Statistical analysis

The preprocessed intensity data were log-transformed ahead of downstream analyses. To control family-wise error rate, the Bonferroni method was used for adjusting $P$-values and the $\alpha$ level was 0.01, unless otherwise specified. The ICC was computed using the method of Shrout and Fleiss (1979) for a set of randomly chosen two raters (Shrout & Fleiss, 1979). The linear association of an antibody signal level with age was tested with ordinary LM using R 3.6.0. The meta-analysis was conducted using the inverse variance method with between-study variance estimated by DerSimonian-Laird model (DerSimonian & Laird, 1986) with "meta 4.9.9" R-package. We used a linear mixed model to address the correlation between twins where the response variable was the normalized antibody measurement and age was a fixed covariate. This model was performed using the R-package "lme4 1.1.21." For the association test for mortality, Cox proportional hazards models were fitted to the survival data with age as the time-scale and right censoring of the age on the updated date of death information (Thiébaut & Bénichou, 2004). In the survival analysis for two group comparison, the subjects in sample set 3 were divided into two groups, top and bottom quarters by the standardized HRG values, which were the scaled residuals of LM where the normalized HRG measurement values were regressed on age and age squared for women and for men separately. The hazard models were adjusted for gender if applicable and for CRP, glucose, or HbA1c as described above. The proportionality assumption of the models was tested using Schoenfeld residuals (Grambsch & Therneau, 1994). Survival analyses, including computation of Harrell's C-index (Harrell et al, 1982), were conducted using the R packages "survival 3.1.8," "eha 2.8.0," and "survminer 0.4.6."

## Data Availability

Researchers interested in using STR data must obtain approval from a Swedish Ethical Review Board and from the Steering Committee of the STR. Researchers using the data are required to follow the terms of an agreement containing a number of clauses designed to ensure protection of privacy and compliance with relevant laws. For further information, contact Patrik Magnusson (Patrik.Magnusson@ki.se).

## Supplementary Information

## Acknowledgements

We like to thank Camilla Björk and Jens Mattsson from MEB at the Karolinska Institutet, everyone in the Affinity Proteomics group at SciLifeLab, and especially Claudia Fredolini, MariaJesus Iglesias, Matilda Dale, Sanna Byström, Martin Zwahlen, Björn Forsström, Björn Winckler, and Philippa Pettingill for supporting this work. We also thank the entire staff of the Human Protein Atlas for their efforts and Hanna Tegel, Johan Rockberg, and their team for providing the recombinant HRG protein. We thank Bernd Wollscheid from ETH Zurich for fruitful discussions about HRG. This work was supported by ProNova VINN Excellence Centre for Protein Technology (VINNOVA, Swedish Governmental Agency for Innovation Systems), the Knut and Alice Wallenberg Foundation for the Human Protein Atlas, Science for Life Laboratory for Plasma Profiling Facility, and Erling Persson Foundation for KTH Centre for Precision Medicine. We acknowledge The Swedish Twin Registry for access to samples and data. The Swedish Twin Registry is managed by Karolinska Institutet and receives funding through the Swedish Research Council under the grant no 2017-00641. LifeGene was supported by grants from the Swedish Research Council, Torsten and Ragnar Söderbergs Foundation, Stockholm County Council, and AFA Försäkringar.

### Author Contributions

M-G Hong: conceptualization, data curation, software, formal analysis, validation, investigation, visualization, methodology, and writing—original draft, review, and editing.
T Dodig-Crnkovic: resources, data curation, formal analysis, validation, investigation, visualization, and writing—original draft, review, and editing.
X Chen: formal analysis, methodology, and writing—original draft, review, and editing.
K Drobin: data curation, investigation, and writing—original draft, review, and editing.
W Lee: formal analysis, methodology, and writing—original draft, review, and editing.
Y Wang: formal analysis and writing—original draft, review, and editing.
F Edfors: data curation, formal analysis, investigation, visualization, and writing—original draft, review, and editing.
D Kotol: data curation, formal analysis, investigation, visualization, and writing—original draft, review, and editing.
CE Thomas: formal analysis and writing—original draft, review, and editing.
R Sjöberg: data curation, investigation, and writing—original draft, review, and editing.
J Odeberg: resources, data curation, and writing—original draft, review, and editing.
A Hamsten: resources, data curation, and writing—original draft, review, and editing.

A Silveira: resources, data curation, and writing—original draft, review, and editing.

P Hall: resources, data curation, and writing—original draft, review, and editing.

P Nilsson: resources and writing—original draft, review, and editing.

Y Pawitan: formal analysis, methodology, and writing—original draft, review, and editing.

M Uhlen: resources, funding acquisition, and writing—original draft, review, and editing.

NL Pedersen: resources, data curation, funding acquisition, and writing—original draft, review, and editing.

S Hägg: formal analysis, validation, investigation, methodology, and writing—original draft, review, and editing.

PKE Magnusson: conceptualization, resources, data curation, formal analysis, funding acquisition, investigation, methodology, and writing—original draft, review, and editing.

JM Schwenk: conceptualization, resources, data curation, formal analysis, supervision, funding acquisition, validation, investigation, methodology, project administration, and writing—original draft, review, and editing.

## Conflict of Interest Statement

M Uhlén is one of the founders of Atlas Antibodies AB, a company that sells Human Protein Atlas antibodies used in this study. P Nilsson, F Edfors and JM Schwenk acknowledge a relationship with Atlas Antibodies AB. The other authors declare no conflict of interest.

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
