## [Reviewer comments · Life Science Alliance]

Life Science Alliance

Profiles of histidine-rich glycoprotein associate with age and risk of all-cause mortality

Mun-Gwan Hong, Tea Dodig-Crnkovic, Xu Chen, Kimi Drobin, Woojoo Lee, Yunzhang Wang, Fredrik Edfors, David Kotol, Cecilia Thomas, Ronald Sjöberg, Jacob Odeberg, Anders Hamsten, Angela Silveira, Per Hall, Peter Nilsson, Yudi Pawitan, Mathias Uhlen, Nancy Pedersen, Sara Hägg, Patrik Magnusson, and Jochen Schwenk

DOI: 10.26508/lsa.202000817

Corresponding author(s): Jochen Schwenk, KTH-Royal Institute of Technology

Review Timeline:

Submission Date:	2020-06-16
Editorial Decision:	2020-06-22
Revision Received:	2020-06-26
Editorial Decision:	2020-07-20
Revision Received:	2020-07-22
Accepted:	2020-07-22

Transaction Report:

Please note that the manuscript was reviewed at Review Commons and these reports were taken into account in the decision-making process at Life Science alliance.

Reviewer #1 (Evidence, reproducibility and clarity (Required)):

****Summary:****

The paper applied affinity based proteomics and antibody validation to choose and validate histidine-rich glycoprotein (HRG) as a protein/target of interest. Survival analysis techniques were used to show associations between this protein and certain biomarkers, age and all cause mortality.

These results and findings were used to conclude that HRG may serve as a molecular indicator of age and mortality risk.

****Major Comments:****

The authors of the paper start the paper with just one protein narrowed down ie. HRG. The rest of the paper uses affinity based proteomics, antibody validation, GWAS and survival analysis to validate this target and support their claim that HRG is an age associate protein linked to mortality and certain clinical outcomes. How did the authors conclude that HRG was the only target to explore further in this paper? What methods or analysis was done for this? What were the other proteins if any that showed up in these studies?

For mortality outcome, it is not clear which class of disease is most strongly associated with increased risk of mortality from elevated HRG levels. If cause-specific mortality exists among the cohorts, could authors provide a more exact breakdown of the type of associated mortality by a disease class?

Page 4 Section 3 (Results)-

The authors say "We found consistent age-associated trends with HPA045005 across all eight replication sets (Supporting Figure 3)". On examining the supporting figure we noticed that the slope for the set with the largest number of subjects (Set 3 with ~3000 people) is visually negligibly positive (showing weakest age associated trends with HPA045005). Some comments from the authors on why they think the largest data set showed the weakest association.

From Figure 2 C in the main manuscript one concludes that for HPA045005, binding for CC individuals is ~ 2 times higher than TT individuals. Is it possible the age association showing up for HPA045005 is primarily a function of changing/increase in allele frequency as a function of age?

The authors could consider adding a clarifying plot of Age vs Allele frequency or adding an interaction term of Age and Allele Frequency in the regression and survival analysis to address this question.

It is interesting that the signals were significant with the HPA045005 antibody but not with the BSI037 antibody. This is in spite of the fact that the GWAS for BSI0137 signals had an even stronger hit to the same locus. Can the authors please comment on why the signals from HPA045005 and BSI0137 were not highly correlated with one another and why the better antibody could not replicate the survival analysis results?

****Minor Comments:****

Figure 1: The authors description of the figure could use more clarification. "For each sample set, the estimated effect from the linear regression model.." estimated effect of what on what? On reading the main text one concludes it is the effect of age on HPA045005. This needs to be clarified in the label.

Figure 3: The X axis for the Kaplan Meir survival curve is labelled as Age. Survival is usually time to event and time is usually the follow up time. Further clarification for the choice of this label might be helpful.

Figure 3: it would be good to include a table with the number of individuals at risk at the bottom of the plot at defined time intervals. The figure currently compares the bottom and top quartiles of HRP for visual assessment of mortality risk, it would also be informative to include middle quantiles.

Supporting Table 5: The note at the bottom of this table states "standardized HRG values by linear regression and scaling." What does standardization by linear regression mean?

Supporting Table 5: It would be useful to understand that HRG carries additional risk beyond known Age and known clinical biomarkers listed in Table 2 (APOA1, APOB, TC, TG, Glucose, LDL). Could authors include a multivariate CoxPH regression with just Age? and with Age + clinical covariates?

Reviewer #1 (Significance (Required)):

The authors have identified a new biomarker for aging and mortality. Understanding the mechanism and pathways involved in HRG homeostasis and how aging causes dysregulation of this HRG could be a topic for further research. Overall, this pathway provides an opportunity of a new molecular target for aging-based drugs and research.

This article should be of interest to researchers interested in the biology of aging and for researchers developing drugs to slow down the process of aging. In addition, it should be of interest to researchers studying the HRG as a biomarker (for example, in sepsis (<https://ccforum.biomedcentral.com/articles/10.1186/s13054-018-2127-5>, https://papers.ssrn.com/sol3/papers.cfm?abstract_id=3437790)).

This paper was reviewed by 3 co-reviewers, a senior principal investigator with extensive bioinformatics, metabolomics/proteomics, epidemiological experience, a highly experienced computational biologist with a record of developing and applying methods in bioinformatics and computational biophysics and lastly an computational biologist with a background in applied mathematics and statistical analysis. All three scientists are interested in aging research and understanding how human physiology and biomarkers in specific, change as a function of age.

Reviewer #2 (Evidence, reproducibility and clarity (Required)):

****Summary****

The manuscript by Hong et al. describes the identification and validation of histidine-rich glycoprotein (HRG) as a marker of chronological age and all-cause mortality. HRG was determined using proteomics of serum and plasma samples in 9 different cohorts (total sample size ~4,100). The association with mortality was tested in the largest available cohort (TwinGene), comprising ~3,000 samples. The association with mortality seems to be stronger in women in comparison to men and could not be explained by CRP or diabetes-related traits. The HRG levels determined using an alternative antibody, BSI0137, did not show any association with mortality, indicating that the effect on mortality is likely isoform-dependent. The performed analyses seem to be statistically solid. However, the association with mortality still needs to be replicated in independent studies and the HRG measurement does not yet seem to be ready for standardized high-throughput measurement, which is necessary to make it usable as biomarker.

****Major comments****

- Although the authors have convincingly identified HRG to be associated with chronological age and mortality, it will require quite some additional work (including replication of the observed association with mortality in independent cohorts, testing the predictive ability, and making the measurement standardized and high-throughput) to prove its use as potential biomarker. At the moment, this is not at all discussed in the manuscript. Moreover, there have been some recent large-scale studies that identified biomarkers at the metabolic level that are not at all mentioned by the authors. The authors only refer once to the recent proteomic study by Lehallier in the Introduction, but do not at all discuss their findings in relation to this paper. Last but not least, HRG has already been associated with mortality in a previous study (<https://www.ncbi.nlm.nih.gov/pubmed/29303798>), but there is no mention of this anywhere in the manuscript. Hence, I think it would be good if the authors perform a thorough literature search to place their findings into context and rewrite their Discussion accordingly.

- The authors need to add a Supplementary Table showing the association of all their 7,258 HPA antibodies with chronological age. Although I trust the authors, I can currently not tell if it is indeed correct that only one antibody was significantly associated with age in set 1.

- According to description in the Supporting Information, several samples in set 3-5 were overlapping with set 1 (45 in total). These samples should be removed from datasets 3-5 to make sure that there are no overlapping samples in the meta-analysis. However, I am not sure if the authors have actually done this. For the GWAS the overlapping samples from set 3 could still be included, given that set 1 is not involved in that. The authors could actually use these 45 overlapping samples to provide additional details about the reproducibility of HPA045005 between different measurements, for example by showing a correlation plot.

****Minor comments****

- When looking at the effect of the rs9898-stratified analysis (Table S2) it seems that there only is an effect in the presence of the C-allele. Have the authors considered the presence of a potential recessive effect of this variant when looking at mortality?

- The authors need to discuss in more detail the implications of the difference between the two HRG antibodies in their association with mortality, for example in light of the use of

HRG levels as a potential biomarker (i.e. how should one deal with the fact the way the levels are measured influences the outcome).

- Why did the authors put part of their Discussion in the Supplement? This is not common practice. They should either move it to the manuscript or remove it completely.

Reviewer #2 (Significance (Required)):

The manuscript is clearly written and the analyses seem to be solid. However, although the findings described in the manuscript are interesting for the ageing field, they only provide a small step in the process of the usability of HRG as biomarker, i.e. many validation and follow-up studies will be necessary to prove its value. There have been some recent biomarker studies that have been much more advanced in this respect, which limits the novelty of this manuscript. I therefore feel that this manuscript may be best suitable for a medium-impact ageing-specific journal.

My fields of expertise are ageing, genetics, and molecular epidemiology. Given my limited expertise when it comes to proteomics, I was not able to provide detailed comments on the methodology concerning this part.

Joris Deelen

Response to reviewer comment for manuscript RC-2020-00207

Reviewer #1 (Evidence, reproducibility and clarity (Required)):

****Major Comments:****

The authors of the paper start the paper with just one protein narrowed down ie. HRG. The rest of the paper uses affinity based proteomics, antibody validation, GWAS and survival analysis to validate this target and support their claim that HRG is an age associate protein linked to mortality and certain clinical outcomes. How did the authors conclude that HRG was the only target to explore further in this paper? What methods or analysis was done for this? What were the other proteins if any that showed up in these studies?

We appreciate this comment which reveals unclear explanation how the protein was chosen for further analysis. The protein profile obtained using HPA045005 was the top and single hit out of 7258 protein profiles using a threshold of adjusted P-value below 0.01. In other words, only the profile of HRG was statistically significantly associated with age in the screening sample set (N = 156). The results of all protein profiles were attached as supplementary Table S1. Phrases about the alpha level were added to the text to make the threshold clear. Because antibody validation of these exploratory studies requires enormous efforts and time, we could not choose a more liberal and inclusive threshold.

For mortality outcome, it is not clear which class of disease is most strongly associated with increased risk of mortality from elevated HRG levels. If cause-specific mortality exists among the cohorts, could authors provide a more exact breakdown of the type of associated mortality by a disease class?

We thank the reviewer for the question and have now added cause-specific data in the manuscript. Using cause of death data, mortality risk by diseases in circulatory system were compared with the risk by neoplasm and others. Elevated HPA045005-HRG profiles were found to associate with mortality risk by diseases of the circulatory system (HR = 1.46 per SD, $P = 2.80 \times 10^{-4}$, ICD-10 code I00-I99). It was larger than the risk by malignant neoplasms (HR = 1.28 per SD, $P = 1.73 \times 10^{-2}$, ICD-10 code C00-C97). We chose big categories as ICD-10 codes "I" and "C" because the number of events was too small to get enough power in the survival analysis.

Page 4 Section 3 (Results)-

The authors say "We found consistent age-associated trends with HPA045005 across all eight replication sets (supplementary Fig S3)". On examining the supporting figure we noticed that the slope for the set with the largest number of subjects (Set 3 with ~3000 people) is visually

negligibly positive (showing weakest age associated trends with HPA045005). Some comments from the authors on why they think the largest data set showed the weakest association.

The plot for each cohort (in supplementary Fig S3) had different ranges in the y-axes. To make those plots comparable, the ranges in the y-axes of the different panels in the figure were modified to be the same for all cohorts. In the new version of the plot, it is easier to notice that there in fact is an increasing trend of the profiles in set 3. As we briefly discussed in Discussion, weaker age-association of the sample set may be due to the set was near to a random sample of population in the age range. Set 1, however, had over-representation of older people by selecting equal number of people in every age-intervals.

From Figure 2 C in the main manuscript one concludes that for HPA045005, binding for CC individuals is ~ 2 times higher than TT individuals. Is it possible the age association showing up for HPA045005 is primarily a function of changing/increase in allele frequency as a function of age?

The authors could consider adding a clarifying plot of Age vs Allele frequency or adding an interaction term of Age and Allele Frequency in the regression and survival analysis to address this question.

As suggested, we now added a test of age association, and average age was compared by genotype. The result was added in supplementary Table S3. The heterozygote (CT) group has slightly higher average age without statistical significance (ANOVA $P = 0.096$).

It is interesting that the signals were significant with the HPA045005 antibody but not with the BSI037 antibody. This is in spite of the fact that the GWAS for BSI0137 signals had an even stronger hit to the same locus. Can the authors please comment on why the signals from HPA045005 and BSI0137 were not highly correlated with one another and why the better antibody could not replicate the survival analysis results?

We thank the reviewer for the comments. We believe that our text about our findings were not clear enough, though it is a primary finding. We modified the main text to easily distinguish the HPA045005-derived profiles that were influenced by the 204th amino-acid of HRG protein, from the BSI0137-derived profiles influenced by the 493th amino-acid. The signals from those two antibodies were likely obtained by capturing different parts of HRG, which are schematically illustrated in Fig 2D. What we found is that only one binder's profiles, not the other's, had predictive power for mortality risk within about 8.5 years. That suggests some age-dependent changes around the 204th residue of HRG reflected biological aging rather than whole protein level. To make our finding clearer, the two binders were compared in Table 2.

****Minor Comments:****

Figure 1: The authors description of the figure could use more clarification. "For each sample set, the estimated effect from the linear regression model.." estimated effect of what on what?

On reading the main text one concludes it is the effect of age on HPA045005. This needs to be clarified in the label.

We agree with the reviewer and have added these words.

Figure 3: The X axis for the Kaplan Meir survival curve is labelled as Age. Survival is usually time to event and time is usually the follow up time. Further clarification for the choice of this label might be helpful.

We clarified the choice of the time scale in the figure legend with a reference, where it was further discussed (Thiébaud & Bénichou, 2004). We chose age as the time scale, seeing age is the strongest risk factor for all-cause mortality, as the suggestion in the reference. We attempted to use follow-up time as the time scale with age adjustment before, which gave us almost the same results but violated the proportionality assumption of COX models.

Figure 3: it would be good to include a table with the number of individuals at risk at the bottom of the plot at defined time intervals. The figure currently compares the bottom and top quartiles of HRP for visual assessment of mortality risk, it would also be informative to include middle quartiles.

The figure was updated accordingly. The risk table was included and the results of the middle group were presented.

supplementary Table S5: The note at the bottom of this table states "standardized HRG values by linear regression and scaling." What does standardization by linear regression mean?

A sentence that explains the standardization was added in the footnote of the table.

supplementary Table S5: It would be useful to understand that HRG carries additional risk beyond known Age and known clinical biomarkers listed in Table 2 (APOA1, APOB, TC, TG, Glucose, LDL). Could authors include a multivariate CoxPH regression with just Age? and with Age + clinical covariates?

The impact of those clinical variables on survival models was examined and the results were added to supplementary Table S6 (which was Table S5). It turned out that the addition of those variables barely changed the results of the model for the HRG profile affected by 202th amino-acid.

Reviewer #2 (Evidence, reproducibility and clarity (Required)):

****Summary****

The manuscript by Hong et al. describes the identification and validation of histidine-rich glycoprotein (HRG) as a marker of chronological age and all-cause mortality. HRG was determined using proteomics of serum and plasma samples in 9 different cohorts (total sample size ~4,100). The association with mortality was tested in the largest available cohort (TwinGene), comprising ~3,000 samples. The association with mortality seems to be stronger in women in comparison to men and could not be explained by CRP or diabetes-related traits. The HRG levels determined using an alternative antibody, BSI0137, did not show any association with mortality, indicating that the effect on mortality is likely isoform-dependent. The performed analyses seem to be statistically solid. However, the association with mortality still needs to be replicated in independent studies and the HRG measurement does not yet seem to be ready for standardized high-throughput measurement, which is necessary to make it usable as biomarker.

****Major comments****

- Although the authors have convincingly identified HRG to be associated with chronological age and mortality, it will require quite some additional work (including replication of the observed association with mortality in independent cohorts, testing the predictive ability, and making the measurement standardized and high-throughput) to prove its use as potential biomarker. At the moment, this is not at all discussed in the manuscript. Moreover, there have been some recent large-scale studies that identified biomarkers at the metabolic level that are not at all mentioned by the authors. The authors only refer once to the recent proteomic study by Lehallier in the Introduction, but do not at all discuss their findings in relation to this paper. Last but not least, HRG has already been associated with mortality in a previous study (<https://www.ncbi.nlm.nih.gov/pubmed/29303798>), but there is no mention of this anywhere in the manuscript. Hence, I think it would be good if the authors perform a thorough literature search to place their findings into context and rewrite their Discussion accordingly.

We appreciate the reviewer's comments on the limitation of our paper. We are aware of the requirement of further investigation on HPA045005-HRG profiles as a biomarker to confirm it with independent cohorts. Instead, we supported our findings with a set of confirmatory analyses; we validated and annotated age-associated profile applying GWAS, sandwich assays, peptide arrays and mass spectrometry. Comparing two antibody profiles, we narrowed down to age-associated region within the protein HRG. The approach and finding, we believe, is novel.

We added some discussion about recent large-scale proteomic studies such as Tanaka et al, 2018 and Lehallier et al, 2019. Unexpectedly, HRG was found not measured in those studies

despite of the protein is one of the abundant proteins in blood (Poon et al, 2011). It may reflect challenges in assay development and missing piece in those large studies. The papers lack further investigation for molecular targets, which is common in proteomic papers, and makes it difficult to compare between studies and technologies. In that sense, our approach is different from other proteomic studies, because we invested time and efforts to investigate the molecular target.

We are though thankful for the introduction of the suggested HRG publication, which we did not know about. We concluded that there are substantial differences in the subjects and suggested functions for the protein. Kuroda et al. found HRG as a biomarker for sepsis of ICU patients, while our study was done on the general population. They were measuring HRG protein level, whereas we found one particular region in HRG as a biomarker for all-cause mortality. Hence, we briefly discussed the reference in the paragraph about general information about HRG.

- The authors need to add a Supplementary Table showing the association of all their 7,258 HPA antibodies with chronological age. Although I trust the authors, I can currently not tell if it is indeed correct that only one antibody was significantly associated with age in set 1.

We agree with the reviewer. The table of association test results of all 7258 antibody profiles was attached to the paper as supplementary Table S1.

We were also surprised that only one passed a conventional P-value threshold 0.01 after Bonferroni correction. It might be due to the low number of samples in the sample set 1 (N=156), compared to the number of antibodies or tests.

- According to description in the Supporting Information, several samples in set 3-5 were overlapping with set 1 (45 in total). These samples should be removed from datasets 3-5 to make sure that there are no overlapping samples in the meta-analysis. However, I am not sure if the authors have actually done this. For the GWAS the overlapping samples from set 3 could still be included, given that set 1 is not involved in that. The authors could actually use these 45 overlapping samples to provide additional details about the reproducibility of HPA045005 between different measurements, for example by showing a correlation plot.

We agree with the reviewer. Those 45 overlapping samples were excluded in the meta-analysis. As the reviewer's comment, only the data of sample set 3 was used for the GWAS.

We also appreciate the comment regarding reproducibility and acknowledge that there are limitations to the technical performance of our exploratory SBA method. The procedure is tailored to handle large number of antibodies and profile 384 sample in the analysis plates. This setup allowed us to process relatively large number of samples per batch but it might be affected by batch effects. In our study set 3, there were 2999 samples randomized and

analyzed in 8 different 384-well plates. The 44 overlapping samples between sets 1 and 3 were added to one of these 8 plates. This resulted in 1-11 samples to be analyzed on the same plate, hence, comparing these 44 with previous assays might be influenced if not dominated by plate effects. We went back to the initial data set generated during 2011/2012 and compared the first data with replicated assays using the same freeze-thawed samples. For HPA045005 we found the data to correlate by $r=0.45$. The next analyses of these 44 samples were conducted during 2015 using different sample aliquots and preparations as well as different SBAs. The correlation to previous assays was $r<0.3$, hence not supportive. However, we acknowledge that there are many influential variables that we were not able to retrospectively decompose or standardize for our exploratory efforts, and we believe that replicating the association in 9 different study sets is still a strong indicator for the validity of the data. In order to provide a more recent measure that can illustrate the reproducibility of SBA assays using HPA045005, we borrow the performance data from our work by Dodig-Crnkovic et al (doi.org/10.1101/2020.03.13.988683). There, we assessed the intra-assay performance of HPA045005 using 12 replicated sample pools of EDTA plasma and determined a %CV < 5%. The inter-assay correlation between replicated rounds of analyses of 68 samples was of $r=0.97$. In essence, time between rounds of analyses, time in freezers as well as further advances in the experimental method should have improved the performance in terms to replication. We acknowledge that coming efforts using this method should consider to include internal standards that might help to track and pinpoint the technical or even sample/protein related differences.

****Minor comments****

- When looking at the effect of the rs9898-stratified analysis (Table S2) it seems that there only is an effect in the presence of the C-allele. Have the authors considered the presence of a potential recessive effect of this variant when looking at mortality?

Average age of the individuals of each genotype of the SNP was compared and added into supplementary Table S3 (which was Table S2). No significant difference between the genotypes was found. As the reviewer noted, the mortality association of the HRG profiles affected by 204th amino-acid in the TT genotype group of rs9898 was milder and did not reach statistical significance. We believe that it is due to substantially smaller sample size and number of deaths in the genetic group. To clarify the difference in numbers, those numbers were added into the supplementary Table S3 (which was Table S2).

- The authors need to discuss in more detail the implications of the difference between the two HRG antibodies in their association with mortality, for example in light of the use of

HRG levels as a potential biomarker (i.e. how should one deal with the fact the way the levels are measured influences the outcome).

We appreciated this valuable comment, which clearly reveals that our claim was not explained sufficiently. We modified the main text to distinguish those two antibody profiles more clearly. We also added Figure 2D and changed the structure of Table 2 to highlight the difference between the two antibody profiles.

- Why did the authors put part of their Discussion in the Supplement? This is not common practice. They should either move it to the manuscript or remove it completely.

We moved the discussion in the supplement to main text as the reviewer's suggestion.

Reviewer #2 (Significance (Required)):

The manuscript is clearly written and the analyses seem to be solid. However, although the findings described in the manuscript are interesting for the ageing field, they only provide a small step in the process of the usability of HRG as biomarker, i.e. many validation and follow-up studies will be necessary to prove its value. There have been some recent biomarker studies that have been much more advanced in this respect, which limits the novelty of this manuscript. I therefore feel that this manuscript may be best suitable for a medium-impact ageing-specific journal.

My fields of expertise are ageing, genetics, and molecular epidemiology. Given my limited expertise when it comes to proteomics, I was not able to provide detailed comments on the methodology concerning this part.

We thank the reviewer for the honest and constructive assessment of our work and agree with the suggestion to transfer this work to a medium-impact journal covering aspects of ageing research.

June 22, 2020

Re: Life Science Alliance manuscript #LSA-2020-00817

Dr. Jochen Schwenk
KTH-Royal Institute of Technology, School of Biotechnology
SciLife Lab Stockholm
Stockholm SE-10044
SWEDEN

Dear Dr. Schwenk,

Thank you for submitting your manuscript entitled "Profiles of circulating histidine-rich glycoprotein associate with age and risk of all-cause mortality" to Life Science Alliance. The manuscript was assessed by expert reviewers at Review Commons and their reports were transferred to us.

In particular, reviewer #2 pointed out that the observed association between HRG and mortality needs to be validated in independent cohorts. While we would not ask you to include additional cohorts at this stage, the limitations in this regard need to be thoroughly discussed. All other issues raised by the reviewers need to be satisfactorily addressed as well.

Thank you for this interesting contribution to Life Science Alliance. We are looking forward to receiving your revised manuscript.

Sincerely,

Reilly Lorenz

Editorial Office Life Science Alliance
Meyerhofstr. 1
69117 Heidelberg, Germany
t +49 6221 8891 414
e contact@life-science-alliance.org
www.life-science-alliance.org

B. MANUSCRIPT ORGANIZATION AND FORMATTING:

July 20, 2020

RE: Life Science Alliance Manuscript #LSA-2020-00817R

Prof. Jochen M. Schwenk
KTH-Royal Institute of Technology
Science for Life Laboratory, Department of Protein Science
Tomtebodavägen 23
Solna SE-17121
Sweden

Dear Dr. Schwenk,

Thank you for submitting your revised manuscript entitled "Profiles of histidine-rich glycoprotein associate with age and risk of all-cause mortality". We would be happy to publish your paper in Life Science Alliance pending final revisions necessary to meet our formatting guidelines.

- please address the remaining reviewer concerns
- please add a callout for Figure S2 to the main manuscript text
- please fix your callouts for panels A&B in Figure S7--these panels are not part of your figure or figure legend
- please provide tables as editable excel or doc files

A. FINAL FILES:

-- Summary blurb (enter in submission system): A short text summarizing in a single sentence the study (max. 200 characters including spaces). This text is used in conjunction with the titles of papers, hence should be informative and complementary to the title. It should describe the context

and significance of the findings for a general readership; it should be written in the present tense and refer to the work in the third person. Author names should not be mentioned.

B. MANUSCRIPT ORGANIZATION AND FORMATTING:

Sincerely,

Reilly Lorenz
Editorial Office Life Science Alliance
Meyerhofstr. 1
69117 Heidelberg, Germany
t +49 6221 8891 414
e contact@life-science-alliance.org
www.life-science-alliance.org

Reviewer #1 (Comments to the Authors (Required)):

The authors have addressed my comments satisfactorily and I feel the manuscript is sufficiently developed for publication in Life Science Alliance.

Reviewer #2 (Comments to the Authors (Required)):

We do not have any additional comments about the paper post our last comments for minor and major revisions. The authors did their best to address many of our concerns. The only one which still seems unresolved to us is the comment/ concern we had about the disconnect between the two antibodies HPA045005 and BSI037. We think the paper could be accepted but with some reservations about the potential non-reproducible signal between the two antibodies.

Response to reviewer comment for manuscript RC-2020-00207R

Reviewer #2:

We do not have any additional comments about the paper post our last comments for minor and major revisions. The authors did their best to address many of our concerns. The only one which still seems unresolved to us is the comment/ concern we had about the disconnect between the two antibodies HPA045005 and BSI037. We think the paper could be accepted but with some reservations about the potential non-reproducible signal between the two antibodies.

We thank the reviewer for the positive assessment of our work and are excited that it is suggested for publication. We do understand that complication arising from the two antibodies and acknowledge that the genetic effects on the circulating proteome is indeed adding a challenge to validate the observations. To acknowledge the remaining concern and the transparently discuss the limitation, we have added the following sentences to the discussion section:

With increasing knowledge about genetic effects on the circulating proteome comes the challenge to validate these observations. As we have seen, genetic variation can cause two antibodies to reveal discordant protein proteins even though they bind the same protein. We acknowledge the limitation in our results to further study the effects on the epitopes of HRG such as by applying additional antibodies and establishing appropriate methods. Such tools would further strengthen the validity and better enable others to reproduce the made observations.

July 22, 2020

RE: Life Science Alliance Manuscript #LSA-2020-00817RR

Prof. Jochen M. Schwenk
KTH-Royal Institute of Technology
Science for Life Laboratory, Department of Protein Science
Tomtebodavägen 23
Solna SE-17121
Sweden

Dear Dr. Schwenk,

Thank you for submitting your Research Article entitled "Profiles of histidine-rich glycoprotein associate with age and risk of all-cause mortality". It is a pleasure to let you know that your manuscript is now accepted for publication in Life Science Alliance. Congratulations on this interesting work.

DISTRIBUTION OF MATERIALS:

Again, congratulations on a very nice paper. I hope you found the review process to be constructive and are pleased with how the manuscript was handled editorially. We look forward to future exciting submissions from your lab.

Sincerely,

Reilly Lorenz
Editorial Office Life Science Alliance
Meyerhofstr. 1
69117 Heidelberg, Germany
t +49 6221 8891 414
e contact@life-science-alliance.org
www.life-science-alliance.org